# The risks of extreme load extrapolation

Stefan F. van Eijk[1], René Bos[1], and Wim A. A. M. Bierbooms[1]

[1]Wind Energy Research Group, Faculty of Aerospace Engineering, Delft University of Technology, 2629 HS Delft, The Netherlands

*Correspondence to:* Stefan van Eijk (sfvaneijk@gmail.com)

**Abstract.** An important problem in wind turbine design is the prediction of the 50-year load, as set by the IEC 61400-1 Design Load Case 1.1. In most cases, designers work with limited simulation budgets and are forced into using extrapolation schemes to obtain the required return level. That this is no easy task is proven by the many studies dedicated to finding the best distribution and fitting method to capture the extreme load behavior as well as possible. However, the issue that is often overloooked is the effect that the sheer uncertainty around the 50-year load has on a design process. In this paper, we use a collection of 96 years' worth of extreme loads to perform a large number of hypothetical design problems. The results show that, even with sample sizes exceeding $N = 10^3$ ten-minute extremes, designs are often falsely rejected or falsely accepted based on an over- or underpredicted 50-year load. Therefore, designers are advised to be critical of the outcome of DLC 1.1 and should be prepared to invest in large sample sizes.

## 1 Introduction

Wind turbine designers are confronted with the IEC 61400-1 Design Load Case 1.1 (IEC, 2005). This evaluates the structural integrity of the major load-carrying components on the basis of a 50-year return level, plus safety factors. As prescribed by Appendix F of the standards, a minimum of 300 minutes of time series—distributed over the relevant wind speeds—will have to be evaluated and followed by an extrapolation scheme to obtain the 50-year return level. Such extrapolations produce notoriously uncertain estimates, which is why the Design Load Case (DLC) 1.1 is often avoided or at least greatly simplified in early stages of the design. However, in cases where DLC 1.1 is design-driving (e.g., for foundations and controllers), dealing with this uncertainty is unavoidable.

Many past efforts to reduce this uncertainty have focused on trying out different sampling methods (Fogle et al., 2008; Agarwal and Manuel, 2009), new modeling techniques (Moriarty et al., 2004; Bos and Veldkamp, 2016), or finding the best distribution type to match the extreme load behavior (Pandey and Sutherland, 2003; Genz et al., 2006; Freudenreich and Argyriadis, 2007; Ragan and Manuel, 2007; Natarajan and Holley, 2008; Peeringa, 2009; Lott and Cheng, 2016). Yet, because most studies deal with relatively small sample sizes (e.g., $\ll 1$ year), the actual uncertainty that surrounds the 50-year return level is often underexposed. With a 63-year data set, Barone et al. (2012b) were able to establish the 90% confidence interval around the 50-year load for sample sizes up to $N = 2,000$ ten-minute maxima. This revealed not only that the 50-year levels are clouded by high uncertainty, but also that they suffer from a considerable bias. Inevitably, this has an effect on the choices made during design.

The aim of this paper is to demonstrate this with a simple exercise, using a collection of 96 years' worth of ten-minute load maxima released by Barone et al. (2012a). The uncertainty distribution is constructed by repeatedly sampling subsets of this data set and obtaining the 50-year loads through an automated extrapolation scheme. We then simulate a problem where a hypothetical designer has to choose between two or more concepts and record how often this uncertainty leads to wrong
choices. The results of this paper should help designers to estimate the required sample sizes for their problem, but also to form a critical attitude concerning the quality and reliability of extrapolated 50-year loads.

## 2   Methodology

Since the focus of this work is on the impact of uncertainty, rather than obtaining the highest possible quality result, the workflow is kept as simple as possible. Loads were extracted by drawing a random sample from a large set of crude Monte
Carlo results and the 50-year return period is found by a graphical fit.

### 2.1   Loads data set

The data set that was used for this study was generated by Barone et al. (2012a). It features the onshore version of the NREL 5 MW reference wind turbine, operating for 96 years in an IEC class 1B climate (IEC, 2005).[1] Ten-minute mean wind speeds were randomly drawn from a Rayleigh distribution, bounded by the cut-in and cut-out wind speeds of 3 and 25 m/s, respectively.
Turbulent wind fields were generated by TurbSim on a $20 \times 20$ grid with a width and height of 137 m and were fed to the FAST v7 aeroelastic code. Every simulation ran for eleven minutes, of which the first minute was discarded to avoid any start-up transients. More details can be found in the original paper.

Each output channel contains over 5 million ten-minute extremes. In this paper, we will use the tower base overturning moment, which plays a major role in the design of foundations. Figure 1 shows the entire set of loads at the respective wind
speeds.

### 2.2   Extrapolation scheme

In many practical situations, a designer does not have the computational resources available to simulate several decades of operation. That is when the 50-year load has to be found by extrapolating.

#### 2.2.1   Aggregation-before-fitting and fitting-before-aggregation

There are several approaches to the extrapolation problem. One method involves drawing a sample directly from the parent mean wind speed distribution. The cumulative distribution of extreme loads then follows naturally from ranking a set of $N$ loads and assigning a plotting position:

$$\hat{F}(M_i) = \frac{i}{N+1}. \tag{1}$$

---

[1]The original paper specifies a class 2B site, but this has been corrected with the release of the data set (see http://energy.sandia.gov/?page_id=13173).

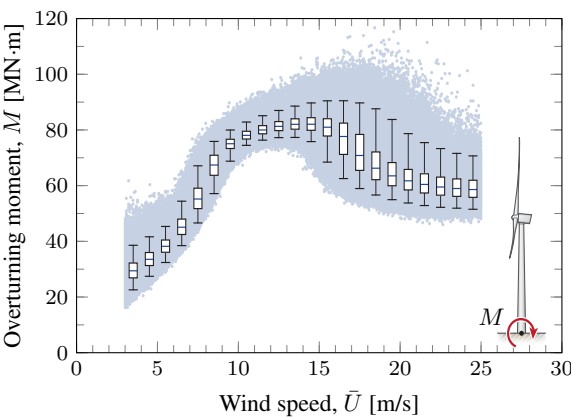

**Figure 1.** The data set, containing over 5 million ten-minute extreme overturning moments between the cut-in and cut-out wind speeds. The box plots indicate the scatter per 1-m/s bin, where the boxes mark the 25th and 75th percentiles, the whiskers mark the 2.5th and 97.5th percentiles, and the bar is the median.

In this case, however, the wind speeds outside of the operating range will have to be accounted for:

$$\hat{F}(M_i) = 1 - \left(1 - \frac{i}{N+1}\right) \int_{\bar{U}_{\text{cut-in}}}^{\bar{U}_{\text{cut-out}}} f(\bar{U}) \, \mathrm{d}\bar{U}, \tag{2}$$

where $f(\bar{U})$ is the mean wind speed distribution. Then, plotting the entire data set yields the return level plot shown in Figure 2. Extrapolation is done by using the entire sample, called *aggregation-before-fitting*. With 96 years' worth of load data, however,

the 50-year return value can be easily matched with a Generalized Extreme Value (GEV) distribution (see next subsection), which yields 115.0 MN·m. Repeating the process by randomly drawing 96-year samples from the same data set allows us to estimate the 95% confidence interval, yielding [113.1, 117.2] MN·m.

Sampling directly from the parent distribution is an example of a *crude Monte Carlo* method, which has the advantage that it gives a raw and unbiased picture of the extreme loads. However, a clear disadvantage is that the bulk of the data originates from

10 relatively low wind speeds where the extremes loads are not expected to lie. In addition, unless a stratified sampling method is used, the wind speeds in any small subsample are not always representative of the parent distribution.

Another method, which is preferred by the IEC guidelines (IEC, 2005), requires the data to be collected in $n$ wind speed bins of a certain width, $\Delta\bar{U}$. Data from every bin is then matched with a distribution function, after which every distribution is weighted according to

15 $$\hat{F}(M) = 1 - \int_{\bar{U}_{\text{cut-in}}}^{\bar{U}_{\text{cut-out}}} f(\bar{U}) \, \mathrm{d}\bar{U} + \sum_{i=1}^{n} F(M|\bar{U}_i) \int_{\bar{U}_i - \frac{1}{2}\Delta\bar{U}}^{\bar{U}_i + \frac{1}{2}\Delta\bar{U}} f(\bar{U}) \, \mathrm{d}\bar{U}. \tag{3}$$

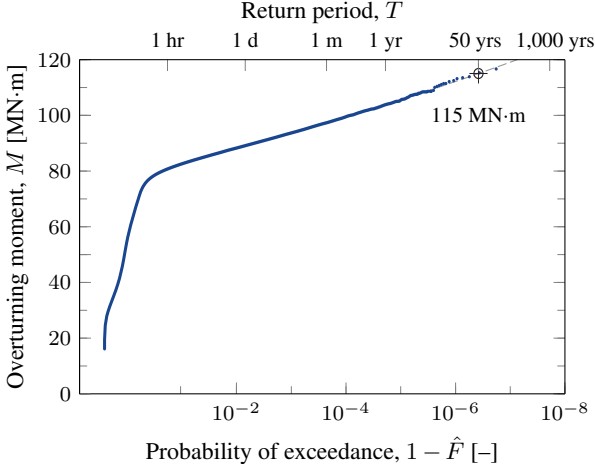

**Figure 2.** Return level plot of the tower base overturning moment, with the entire 96-year data set (i.e., aggregation-before-fitting). A GEV fit above the threshold given by Equation (6) yields a 50-year value of 115.0 MN·m.

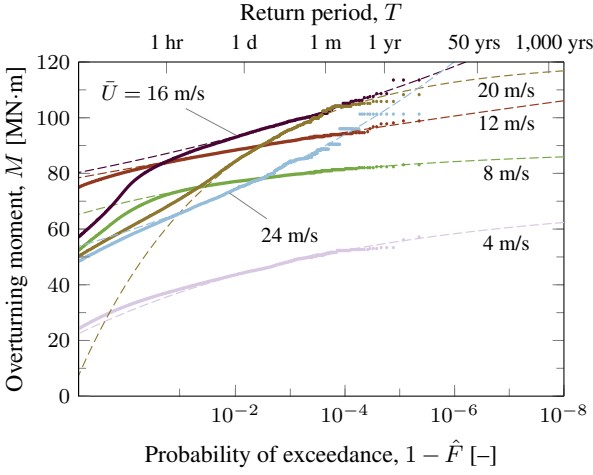

**Figure 3.** Return level plot of some 1-m/s wide bins with a GEV distribution fit after a threshold given by Equation (6).

This is called *fitting-before-aggregation*. It has the advantage that the data in each wind speed bin has less variation and matches closer to an underlying distribution. However, the obvious disadvantage is that a factor $n$ fewer data points are available in each bin to fit (e.g., see Figure 3 and 4).

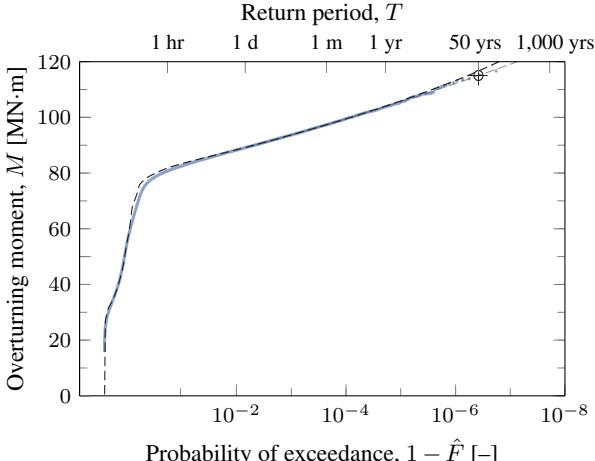

**Figure 4.** Return level plot of the tower base overturning moment by a weighted sum of the bins shown in Figure 3 (i.e., fitting-before-aggregation), using an equivalent 96-year sample size. The weighted distribution is given by a black dashed line, and is overlaid on Figure 2 for comparison.

### 2.2.2 Choice of distribution function

The tail behavior is matched with a distribution function, for example by least-squares fitting. A good candidate for this is the generalized extreme value distribution (GEV):

$$G(M; \mu, \sigma, \xi) = \exp\left[-\left(1 + \xi\frac{M-\mu}{\sigma}\right)^{-1/\xi}\right], \tag{4}$$

where $\mu$ is the location parameter, $\sigma$ the scale parameter, and $\xi$ the shape parameter. A possible alternative is to fix $\xi = 0$, which produces a two-parameter Gumbel distribution:

$$G(M; \mu, \sigma) = \exp\left[-\exp\left(-\frac{M-\mu}{\sigma}\right)\right]. \tag{5}$$

A Gumbel distribution appears as a perfectly straight line on *Gumbel paper* (i.e., on a double-logarithmic scale as in Figure 2) and is often a good first guess of the tail behavior.

### 2.2.3 Identifying the distribution tail

The tail of the distribution shows a characteristic bend, or "knee", that hints that more than one process is at work. Indeed, tracing back the wind speeds belonging to the 10% highest loads points towards a region well above the rated wind speed (see Figure 5). It turns out that this is due to a particular controller response to negative gust amplitudes (e.g., Bos et al., 2015; Bos and Veldkamp, 2016), which also explains the shape of the scatter plot in Figure 1.

To estimate the uncertainty that comes from repeatedly extrapolating different sets of loads, this process has to be automated. However, the difficult part is then to decide where the tail exactly starts under varying sample size. A simple solution that seems

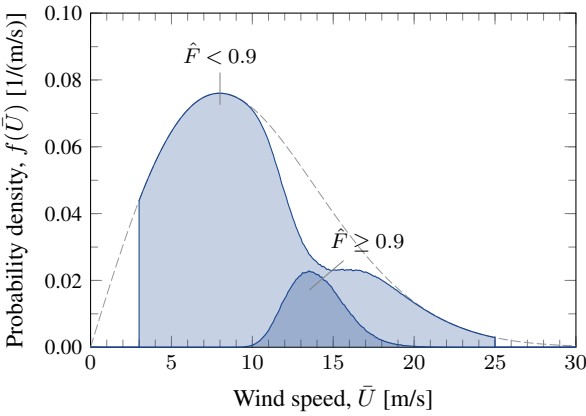

**Figure 5.** Histogram of the sampled wind speeds, where the dashed line marks the Rayleigh mean wind speed distribution belonging to an IEC class 1B climate. The light and dark filled areas correspond to the lowest 90% and highest 10% loads, respectively.

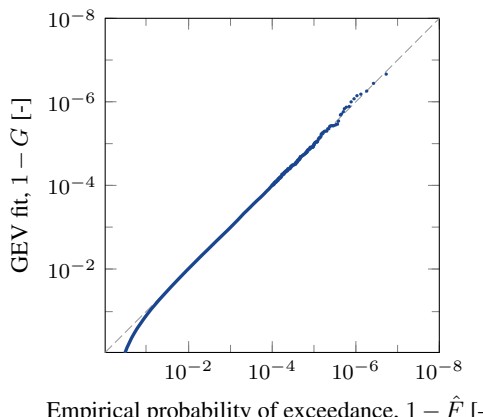

**Figure 6.** Q-Q plot, showing how well the empirical tail behavior matches with an generalized extreme value distribution ($\mu = 78.2$ MN·m, $\sigma = 2.05$ MN·m, $\xi = 0.026$).

to work in most cases is to assume that the tail covers the second half of the distribution when drawn on Gumbel paper; i.e., above a threshold

$$-\ln\left[-\ln\left[\hat{F}\right]\right] > -\tfrac{1}{2}\ln\left[-\ln\left[\hat{F}(M_1)\right]\right] - \tfrac{1}{2}\ln\left[-\ln\left[\hat{F}(M_N)\right]\right]. \tag{6}$$

For the full data set, this means that a distribution is fitted to the upper 0.07% of the data. For a GEV fit, this results in the Q-Q plot shown in Figure 6.

**Table 1.** Extrapolation cases.

| | Sampling method (*parent distribution*) | Distribution fit | Fitted only above threshold |
|---|---|---|---|
| (a) | Aggregation-before-fitting (*Rayleigh*) | Gumbel | No |
| (b) | Aggregation-before-fitting (*Rayleigh*) | Gumbel | Yes |
| (c) | Aggregation-before-fitting (*Rayleigh*) | GEV | No |
| (d) | Aggregation-before-fitting (*Rayleigh*) | GEV | Yes |
| (e) | Fitting-before-aggregation (*Uniform*) | Gumbel | No |
| (f) | Fitting-before-aggregation (*Uniform*) | Gumbel | Yes |
| (g) | Fitting-before-aggregation (*Uniform*) | GEV | No |
| (h) | Fitting-before-aggregation (*Uniform*) | GEV | Yes |

## 2.3 Workflow

The different approaches discussed in the previous subsection are used to set up eight extrapolation cases (see Table 1). The sampling method describes whether the data is drawn directly from its parent Rayleigh distribution and fitted to the entire set (*aggregation-before-fitting*), or if it is constructed from 1-m/s wide bins with an equal number of data points per bin

5 (*fitting-before-aggregation*). Extrapolation is done by matching the data points with either a *Gumbel* or *GEV* distribution by least-squares fit. The final column states whether the entire sample is used (*No*) or only the points that lie above the threshold given by Equation (6) (*Yes*).

These cases are automated for $k = 10^3$ sets of loads, which yields a collection of 50-year return levels, distributed according to $f(\hat{M}_{50\,\text{yrs}})$:

$$
\begin{array}{c}
\quad\quad rank \rightarrow \\[6pt]
M_{1,1} \leq M_{2,1} \leq \ldots \leq M_{N,1} \rightarrow \hat{M}_{50\,\text{yrs},1} \\[6pt]
M_{1,2} \leq M_{2,2} \leq \ldots \leq M_{N,2} \rightarrow \hat{M}_{50\,\text{yrs},2} \\[6pt]
\vdots \quad\quad \vdots \quad\quad\quad\quad \vdots \\[6pt]
M_{1,j} \leq M_{2,j} \leq \ldots \leq M_{N,j} \rightarrow \hat{M}_{50\,\text{yrs},j} \\[6pt]
\vdots \quad\quad \vdots \quad\quad\quad\quad \vdots \\[6pt]
M_{1,k} \leq M_{2,k} \leq \ldots \leq M_{N,k} \rightarrow \hat{M}_{50\,\text{yrs},k} \\[6pt]
\downarrow \\[6pt]
f(\hat{M}_{50\,\text{yrs}})
\end{array}
$$

($\downarrow set$)

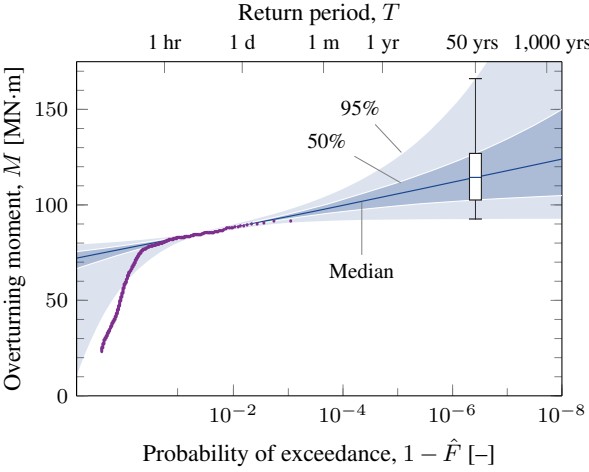

**Figure 7.** Return level plot of the tower base overturning moment, showing all the $k = 10^3$ GEV fits with a sample size of $N = 10^3$. The dot markers belong to one of the $10^3$ samples.

The medians and other quantiles are then estimated by sorting. For example in the case #4 (the crude Monte Carlo method with a GEV fitted above a threshold), the end result is the situation depicted in Figure 7, which can be repeated for different sample sizes.

## 3   Results

Based on the load set and the extrapolation scheme, we can estimate how far a single 50-year load prediction would be off from the true value.

### 3.1   Uncertainty surrounding the 50-year overturning moment

Figure 8 shows how the median and confidence intervals around the 50-year level vary as a function of sample size, $N$. Evidently, the larger the sample size, the smaller the error. In addition, there are some interesting differences between the 8
approaches. Comparing the left column (a, c, e, g) to the right column (b, d, f, h), it seems that a threshold is indeed needed to establish a reliable fit of the distibution tail. Even when the data point are sampled from 1-m/s wide bins, the distributions are often bent and hardly match with any single distribution. The extreme case is when the full empirical load distribution of the aggregation-before-fitting approach (e.g., see Figure 2) is fitted to a Gumbel (a), producing errors well in excess of +100%. In the case of the GEV (c), the fit often takes on strongly negative values of $\xi$. This leads to a reversed Weibull distribution with
an upper bound, which produces a negative bias.

The fitting-before-aggregation approach (e–h) tends to suffer from a positive bias. Likely, this is because most of the partial distributions have a slight downwardly curved tail. Such a shape requires a large enough sample size to fully establish. Small sample sizes, on the other hand, tend to result in a fit with a too large slope that overpredicts the 50-year load.

The tail of the full data set has a slight upwardly curved tail that matches best to a GEV distribution with a small positive $\xi$ (see Figures 2 and 6). However, the Gumbel distribution is clearly more forgiving at small sample sizes. A fixed $\xi = 0$ has the advantage that the fit always stays close to the ideal value, which is especially helpful if the tail of the empirical distribution only contains a few data points.

In addition, the root-mean-squared (RMS) error provides a single measure for the quality of the result:

$$\varepsilon_{\mathrm{RMS}} = \sqrt{\frac{1}{k} \sum_{j=1}^{k} \left( \hat{M}_{50\,\mathrm{yrs},j} - M_{50\,\mathrm{yrs}} \right)^2}, \tag{7}$$

where $M_{50\,\mathrm{yrs}}$ is the "true" 50-year level. As shown in Figure 9, the aggregation-before-fitting approach with a Gumbel fitted above a threshold (b) produces the lowest RMS error. Most of the other approaches show a clear improvement with increasing sample size. Ultimately, the RMS error of (d) falls into the classic $1/\sqrt{N}$ rule that is often found with Monte Carlo methods.

Of course, there are many other approaches to the extrapolation problem that lead to different quality results. In this paper, however, the focus is more on demonstrating how these errors affect a design process. Out of the 8 approaches presented here, 2 are selected. The first is the aggregation-before-fitting approach with a GEV fit above a threshold (d), which has a relatively small bias but a large spread. The second is the fitting-before-aggregation approach with a Gumbel fitted above a threshold (f), which has a large bias but a smaller spread.

## 3.2 Effect on decision making

How this uncertainty affects the decision making process is demonstrated here with a very simple example, where the choice between 2 designs is based on a stress level. Say that a new concept is proposed that is an exact copy of the original NREL 5 MW machine, but with a different wall thickness at the base of the tower. The second moment of area then changes according to

$$I_{yy} = \frac{\pi}{4} \left[ r^4 - (r-t)^4 \right], \tag{8}$$

where $r = 3$ m is the base radius and $t = 35$ mm is the original wall thickness (Jonkman et al., 2009). An extreme overturning moment would cause a compressive stress of

$$\sigma_z = \frac{Mr}{I_{yy}} + \frac{mg}{A}, \tag{9}$$

where $mg = 6.82$ MN is the total weight of the wind turbine and

$$A = \pi \left[ r^2 - (r-t)^2 \right], \tag{10}$$

is the cross-sectional area of the tower base section.

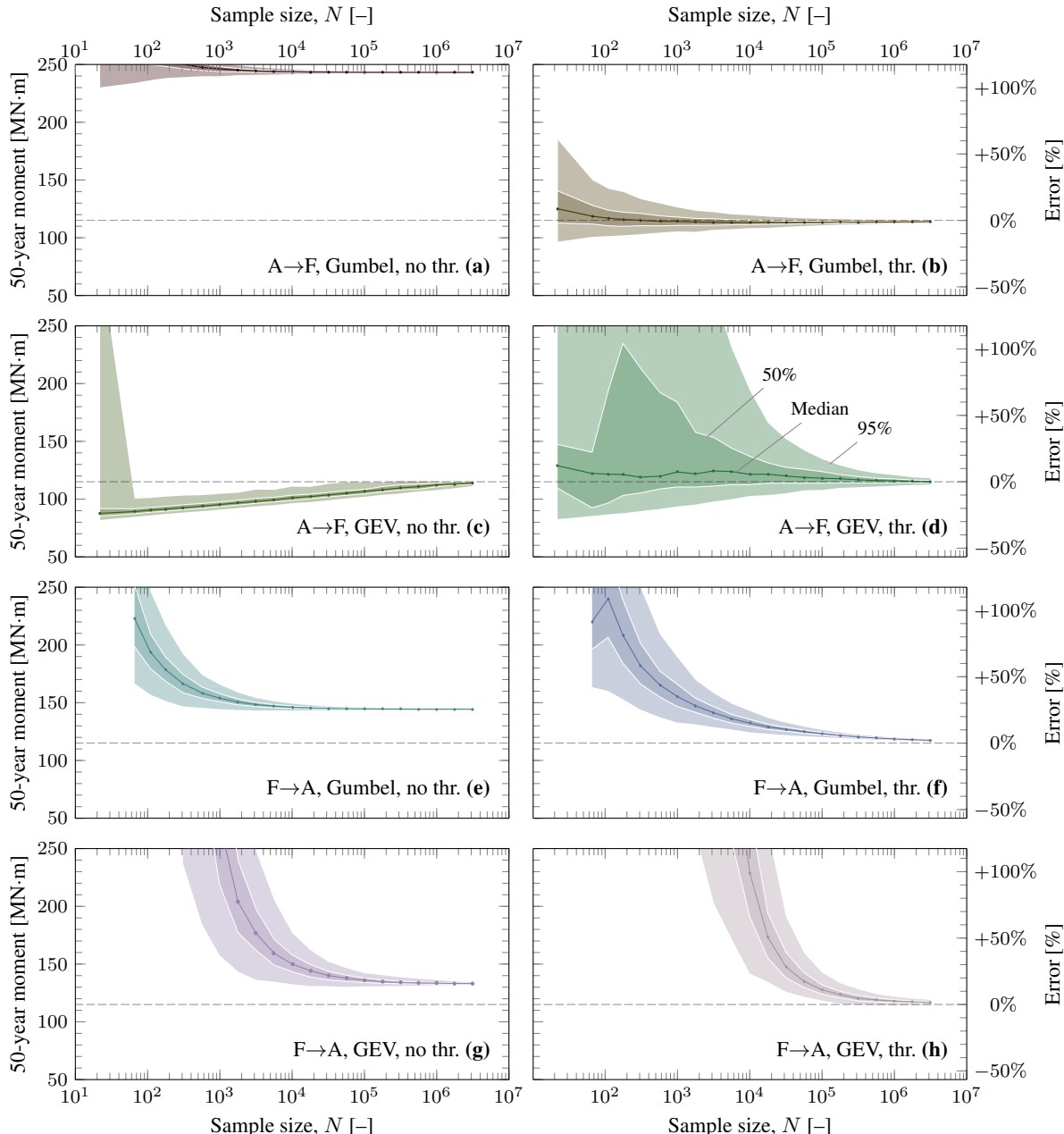

**Figure 8.** Error in the extrapolated 50-year overturning moment, as a function of sample size.

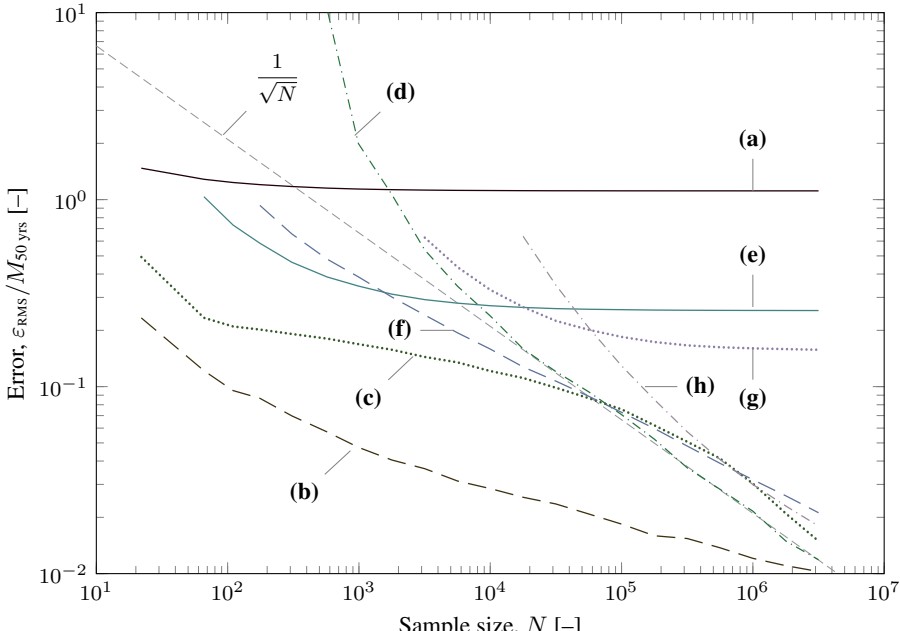

**Figure 9.** Root-mean-squared error in the extrapolated 50-year overturning moment, as a function of sample size.

The objective of the exercise is to find a new wall thickness to reduce the 50-year stress levels; i.e.,

$$\sigma_{z,\text{new}} < \sigma_{z,\text{old}}. \tag{11}$$

This might seem trivial at this point—any thicker wall is guaranteed to reduce the stresses—but the actual difficulty is to determine the 50-year moment. Whereas the original design has already gone through an extensive load analysis from which
the 50-year load level is known, any new concept has to go through this process again.[2]

Due to the uncertainty that surrounds this 50-year level, the new design can be falsely rejected or falsely accepted. Figure 10, for example, shows how often this happens when the load analysis is carried out with the aggregation-before-fitting approach and a sample size of $N = 10^3$. When the wall thickness is reduced by 10% to 31.5 mm, the new design will appear to have lower stresses in 18% of the cases (i.e., the *false positives*). On the other hand, even when the wall thickness is increased by
10% to 38.5 mm, the new design has a 47% chance to still be rejected (i.e., the *false negatives*).

The closer a new design is to the original, the larger the required sample size (see Figure 11). In the case of a large positive bias (see Figure 12), a new concept is nearly always rejected, even if it has exactly the same thickness as the original.

Another case is a comparison between several concepts, where the 50-year stress levels contain the same degree of uncertainty. Five concepts, from 25- to 45-mm wall thickness, are ranked among each other, such that

$$\sigma_{z,1} \leq \sigma_{z,2} \leq \sigma_{z,3} \leq \sigma_{z,4} \leq \sigma_{z,5}. \tag{12}$$

---

[2]Maybe not for something like a new wall thickness, but more so for different control schemes or for rigorous changes to the blade design.

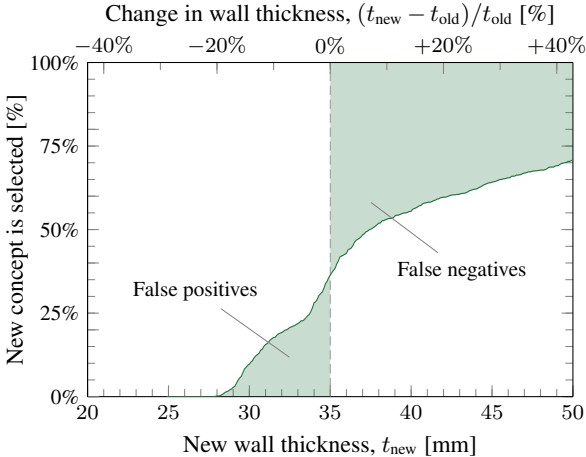

**Figure 10.** Outcome of a concept selection, where a new wall thickness if either accepted or rejected on the basis of a 50-year stress level using an aggregation-before-fitting approach with a GEV fitted above a threshold with a sample size of $N = 10^3$.

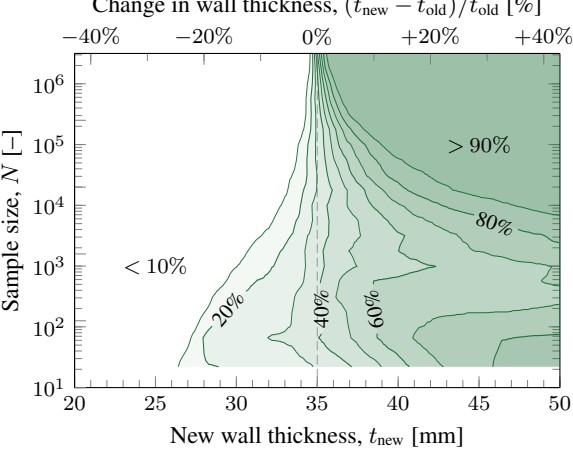

**Figure 11.** Relative number of times a new wall thickness is accepted on the basis of a 50-year stress level, predicted with an aggregation-before-fitting approach with a GEV fitted above a threshold.

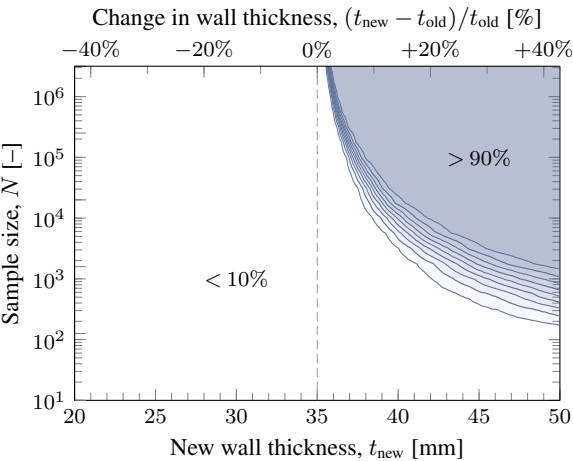

**Figure 12.** Relative number of times a new wall thickness is accepted on the basis of a 50-year stress level, predicted with a fitting-before-aggregation approach with a Gumbel fitted above a threshold.

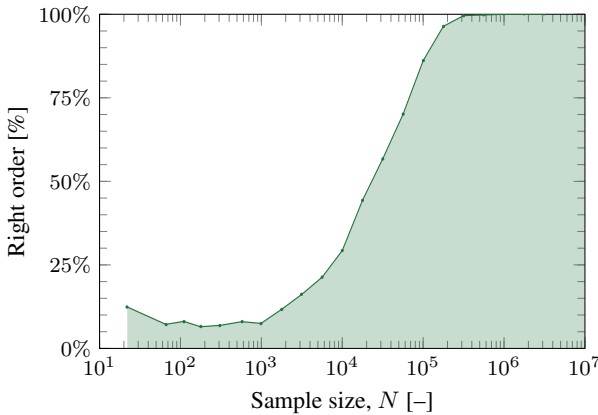

**Figure 13.** How often five concepts, ranging from 25- to 45-mm wall thickness, are ranked in the right order from lowest stress to highest using an aggregation-before-fitting approach with a GEV fitted above a threshold.

In the ideal case, the 45-mm wall thickness should end up at rank 1, the 40-mm one at rank 2, etc. However, how often this ideal ranking happens in practice is shown in Figures 13 and 14. This is where a small spread is preferred over a small bias. As long as the concepts are close to the same mean value, they can still be effectively compared. After $N = 10^4$, the uncertainty is small enough for the order to be right roughly 100% of the time in the case of fitting-before-aggregation (Figure 14). For aggregation-before-fitting (Figure 13), this is not until $N = 3 \cdot 10^5$.

How often each rank is assigned to each concept is shown in Figures 15 and 16. Clearly, the 45-mm wall thickness does not always appear the best and the 25-mm wall thickness does not always appear the worst. In fact, there are cases where the 45-mm wall thickness ends up being the worst of the 5 concepts (see Figure 15 for $N = 10^2$).

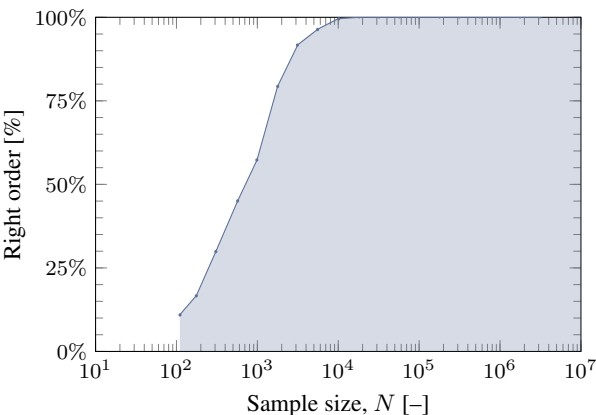

**Figure 14.** How often five concepts, ranging from 25- to 45-mm wall thickness, are ranked in the right order from lowest stress to highest using a fitting-before-aggregation approach with a Gumbel fitted above a threshold.

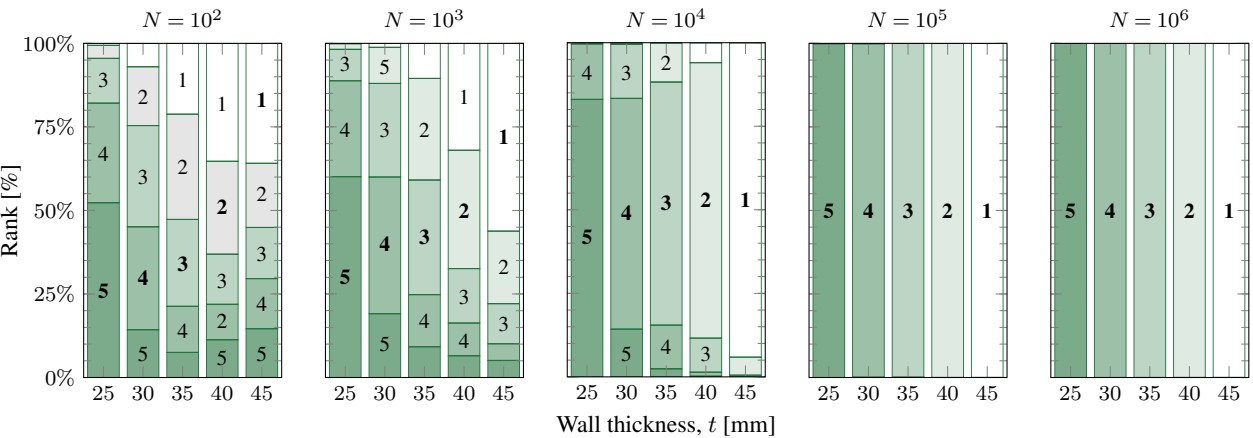

**Figure 15.** Ranking of five wall thicknesses on the basis of a 50-year stress level using an aggregation-before-fitting approach with a GEV fitted above a threshold with different sample sizes.

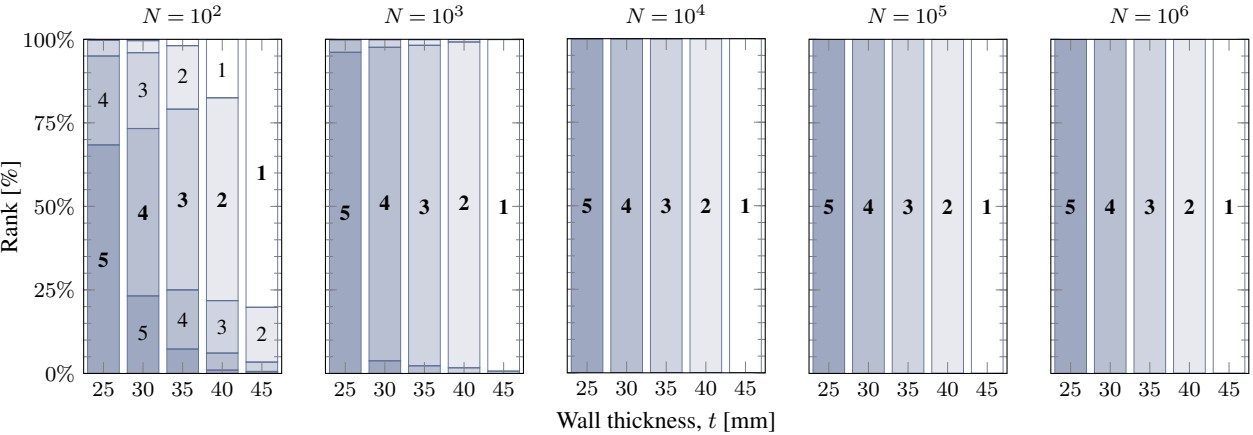

**Figure 16.** Ranking of five wall thicknesses on the basis of a 50-year stress level using a fitting-before-aggregation approach with a Gumbel fitted above a threshold with different sample sizes.

## 4 Discussion

The uncertainty around the 50-year level clearly has a very large impact on decision making. In this paper, we have focused on wall thickness in order to produce results that are counter-intuitive. This is to demonstrate that extrapolated 50-year values can be misleading and can easily trick the designer into making bad choices. In this case, the bad choices are obvious. However,

they can be very difficult to spot in many other cases. For example, when choosing between different foundation types, tuning the gain settings for a controller, or even deciding whether DLC 1.1 is driving over other IEC load cases. The positive bias that is often present in extreme load extrapolations (e.g., Barone et al., 2012b; Van Eijk, 2016) makes it particularly difficult to prove that new designs are capable to reduce 50-year levels. However, it will take an immense computational effort to completely remove the uncertainty from the design process. It is therefore very important that the designer is skeptical enough of their own

results.

Situations where good designs are wrongly discarded or where bad designs are wrongly accepted have a high chance of occurring when the sample sizes are small, especially during the initial design phases. In any case, we can conclude that the bare minimum of 300 minutes of time series, as prescribed in Appendix F of the standards (IEC, 2005), is not sufficient to produce any reasonable 50-year estimate (at least not when using one of the 8 approaches in this paper). One should instead

aim for sample sizes larger than $N = 10^4$, and preferably larger than $N = 10^5$. The effects of changes in wall thickness that in the order of more than 10% are then easily recognizable.

The most obvious solution to reduce the uncertainty is to use high-performance computers in order to run extensive simulation campaigns (e.g., Barone et al., 2012a, b). An alternative remedy is to rely on importance sampling, which is a well-known variance reduction method that allows the user to allocate the computational resources for the most severe conditions (e.g., Bos

et al., 2015; Bos and Veldkamp, 2016).

## 5    Conclusions

The goal of this paper was to demonstrate the effects of the uncertainty around extrapolated 50-year loads. It showed that, unless very large sample sizes are used, DLC 1.1 is a very unreliable measure for the performance of a design. This uncertainty has a pronounced effect on early phases of the design, when computational resources are often scarce.

One should always take into account that it is very time-consuming to prove that concepts are able to reduce the 50-year load, unless the design changes are very radical. In one example using an aggregation-before-fitting approach, where the bottom tower wall thickness of the NREL 5 MW reference turbine was varied, a 10% increase in wall thickness was identified as a way to reduce the stress in only 53% of the cases with a sample size of $N = 10^3$. In fact, more than $10^5$ simulations were required to decrease the probability of a false rejection to 10%. Using a fitting-before-aggregation approach instead led to a strong positive bias and a rejection of the new concept in most cases.

These results show that a critical attitude is required when judging extrapolated extreme loads. When DLC 1.1 is not the design-driver, it might be best to avoid it altogether in early phases of the design. Otherwise, using high-performance computing or importance sampling methods will be the best approach.

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
