# Peer review of "The risks of extreme load extrapolation"

_Wind Energy Science, 2017_

## Referee Comment (RC1) · N. Dimitrov (Referee) · 14 Apr 2017

**The risks of extreme load extrapolation**

The authors present a study on the uncertainty inherent to statistical load extrapolation due to the limited sample sizes. The article is well written and easy to follow. However, there are several issues which are outlined in the following comments:

**General comments**

1) The distribution fit to a random process is representative only if the process is stationary and ergodic (i.e., its statistical properties can be sufficiently well determined over a given sampling period). Subsequently stationarity and ergodicity are central requirements for the validity of load extrapolations. In the present paper, these conditions are not taken into account and I am afraid that they are violated in many of the cases. As an example, for samples drawn from a Rayleigh distribution with mean $\mu$, the standard deviation of the sample mean is defined as $\sigma_{\mu_s} = \mu\sqrt{(4 - \pi)/(N\pi)}$, where $N$ is the sample size. For $\mu = 8.5 m/s$ as in the present paper and for sample size $N = 1000$, the standard deviation of the sample mean wind speed will be 0.14m/s, which means that the 95% confidence interval will be 0.55m/s wide. In effect, samples of this size may be quite different from each other and will result in different extrapolations. This sampling uncertainty arises due to the sample size being too small to properly represent the statistical properties of the parent distribution, and not due to the quality of any subsequent distribution fit. The quality of the fit should relate instead to the realization-to-realization uncertainty for samples drawn from the same stationary process. To summarize – the uncertainty sources should be distinguished and their effect accounted for separately, or the sampling statistical uncertainty should be eliminated by drawing samples with the same wind statistics.

2) The extrapolation procedure analysed by the authors is not necessarily consistent with the actual design approach used by manufacturers. According to the IEC 61400-1 standard, it is acceptable to define the long-term distribution of loads according to two approaches:
1) first carry out extrapolations for simulations binned according to wind speed, and then aggregate the long-term load distribution based on the extrapolated functions (so-called "extrapolate, then aggregate" approach).
2) "First aggregate, then fit on the aggregated distribution" – i.e. the approach used by the authors where realizations are drawn directly from the long-term wind distribution.
In practice, approach 1) is much more common because it requires fewer simulations (approximately 300-400 10-minute periods), it allows using the same simulations used in calculating the fatigue limit state (DLC1.2), and the simulation sample is not conditional on the wind speed distribution. In fact, approach 2) is not even included in the IEC61400-1, ed.3 version cited by the authors – it first appears in the amendment to the standard published in 2010.

3) There is no discussion on the effect of the variation of the wind speed in the original MC sample, and in the random subsets used to test the uncertainty in extrapolation techniques. This variation means that in each sample there are a lot more simulations at wind speeds around the mean, but only few at high wind speeds – e.g., for a sample of size 1000, only 13 simulations on average will represent wind speeds above 20m/s.

4) What about other load cases? Very often, if the design loads from one load case (e.g. DLC1.1) are low, another load case (e.g. emergency shutdown with gust, DLC4.2) will become design-driving. So a lower design load prediction in DLC1.1 will not necessarily lead to lower material thickness, it will simply eliminate the load case as a potential design driver.

5) Page 10, lines 16-20, the authors conclude: "In any case, we can conclude that the bare minimum of 300 minutes of time series, as prescribed in Appendix F of the standards (IEC, 2005), is not sufficient to produce any reasonable 50-year estimate. Based on this load set and this extrapolation produce, one should instead aim for sample sizes larger than $N = 10^5$." I do not agree with this conclusion. Due to the issues outlined in my comments above, some additional uncertainty is present in the extrapolation. If these uncertainties are eliminated, or if another extrapolation procedure (e.g. aggregation after fitting) is used, the accuracy of the extrapolation may improve and the required sample size will become smaller.

**Specific comments**:

6) Page 3, last paragraph: "the wind speeds belonging to the 10% highest loads points towards a region well above the rated wind speed (see Figure 3)". I don't think this statement agrees with Figure 3, where the mode of the distribution $f\left(\overline{U}\middle|\hat{F} \geq 0.9\right)$ is actually at wind speeds just below rated.

7) Page 4, line 5: Why is the GEV distribution a good candidate? In my experience, the GEV, or other 3-parameter representation as the 3-parameter Weibull, are good candidates for extrapolation in situations where only few data points are available and not all of them belong to the upper tail of the extremes distribution. However, if the amount of data and the threshold selection result in a data set which is predominantly from the upper tail which has a characteristic log-linear behaviour, I would think that a 2-parameter Gumbel distribution would provide more robust and accurate fit.

8) Page 5, line 5: "any extrapolated 50-year loads that are more than 50% higher than the "real" value are discarded and resampled": - This data censoring approach is quite crude. It also relies on the known true value which will not be known in practice. Given the large number of samples, it is possible to establish a confidence interval for each load level and discard the outliers, see e.g. Naess and Gaidai, Structural Safety 31 (2009), pp. 325-334.

9) Figure 9: It seems that the uncertainty for sample size N = 100 is smaller than the uncertainty for N = 1000. This is against the logic and indicates a possible sampling bias. Could the authors find an explanation for this?

**Technical comments**:

10) Page 8, lines 5-10: It does not get entirely clear how the stress criteria for accepting or rejecting a given design relate to the extrapolated loads. Please improve the explanation.

---

## Referee Comment (RC2) · Anonymous Referee #2 · 26 Apr 2017

The paper is clear and well-written. It addresses an important practical component of wind turbine design, namely, the extrapolation of loads to a 50-year return value as described in IEC-61400 DLC 1.1. The authors use a simple but effective example of how uncertainties associated with sample size can lead to poor design decisions. The authors should consider the following comments:

1. What is the procedure for obtaining the 50-year load from the data? There is some uncertainty associated with this value, and the authors should attempt to estimate it (for example, from a bootstrap procedure).

2. It is not clear how the results might differ should one choose a different distribution other than GEV for the fit. While an exhaustive study of many possible distributions is not required, some discussion and exploration of this is in order.

---

## Author Comment (AC1) · 3 May 2017

Thank you for the helpful comments; they are much appreciated. We have prepared a new version that hopefully addresses the open issues.

**General comments:**

1. The distribution fit to a random process is representative only if the process is stationary and ergodic (i.e., its statistical properties can be sufficiently well determined over a given sampling period). Subsequently stationarity and ergodicity are central requirements for the validity of load extrapolations. In the present paper, these conditions are not taken into account and I am afraid that they are violated in many of the cases. As an example, for samples drawn from a Rayleigh distribution with mean $\mu$, the standard deviation of the sample mean is defined as $\sigma_{\mu_s} = \mu\sqrt{(4-\pi)/(N\pi)}$, where $N$ is the sample size. For $\mu$ = 8.5 m/s as in the present paper and for sample size $N$ = 1000, the standard deviation of the sample mean wind speed will be 0.14 m/s, which means that the 95% confidence interval will be 0.55 m/s wide. In effect, samples of this size may be quite different from each other and will result in different extrapolations. This sampling uncertainty arises due to the sample size being too small to properly represent the statistical properties of the parent distribution, and not due to the quality of any subsequent distribution fit. The quality of the fit should relate instead to the realization-to-realization uncertainty for samples drawn from the same stationary process. To summarize – the uncertainty sources should be distinguished and their effect accounted for separately, or the sampling statistical uncertainty should be eliminated by drawing samples with the same wind statistics.

   The uncertainty arising from the sampling error is exactly what we wanted to highlight. Perhaps Figure 5 gave the impression that the error was computed by fitting many distributions to a single sample. We have clarified that in the caption.

   We also did not want to go into too much detail here, especially since the focus was on the consequences of the uncertainty rather than on the statistical method itself. Our goal with this paper was to come up with an extrapolation method that was easy to follow for readers struggling with statistical methods, and to challenge them with the consequences of the uncertainty through a simple example. That being said, we understand that perhaps presenting alternative approaches to ours might improve the paper. In the new version, we have gone through the following 8 workflows:

|     | Sampling method (distribution)              | Distribution fit | Fitted only above threshold |
| --- | ------------------------------------------- | ---------------- | --------------------------- |
| (a) | Aggregation-before-fitting (Rayleigh)       | Gumbel           | No                          |
| (b) | Aggregation-before-fitting (Rayleigh)       | Gumbel           | Yes                         |
| (c) | Aggregation-before-fitting (Rayleigh)       | GEV              | No                          |
| (d) | Aggregation-before-fitting (Rayleigh)       | GEV              | Yes                         |
| (e) | Fitting-before-aggregation (Uniform)        | Gumbel           | No                          |
| (f) | Fitting-before-aggregation (Uniform)        | Gumbel           | Yes                         |
| (g) | Fitting-before-aggregation (Uniform)        | GEV              | No                          |
| (h) | Fitting-before-aggregation (Uniform)        | GEV              | Yes                         |

2. The extrapolation procedure analysed by the authors is not necessarily consistent with the actual design approach used by manufacturers. According to the IEC 61400-1 standard, it is acceptable to define the long-term distribution of loads according to two approaches: 1) first carry out extrapolations for simulations binned according to wind speed, and then aggregate the long-term load distribution based on the extrapolated functions (so-called "extrapolate, then aggregate" approach).

In the original manuscript, we decided to go with approach 2) to match the workflow with which the database was generated (Barone et al., 2012). In the new version, we have also included a fitting-before-aggregation approach to be more consistent with IEC guidelines (see point 1).

3. There is no discussion on the effect of the variation of the wind speed in the original MC sample, and in the random subsets used to test the uncertainty in extrapolation techniques. This variation means that in each sample there are a lot more simulations at wind speeds around the mean, but only few at high wind speeds – e.g., for a sample of size 1000, only 13 simulations on average will represent wind speeds above 20m/s.

The variation in the original MC sample (from Sandia) should be clear from Figures 1 and 2. We have also added a small remark on the variation of wind speeds in any small subsample and how it compares to other sampling methods.

4. What about other load cases? Very often, if the design loads from one load case (e.g. DLC1.1) are low, another load case (e.g. emergency shutdown with gust, DLC4.2) will become design-driving. So a lower design load prediction in DLC1.1 will not necessarily lead to lower material thickness, it will simply eliminate the load case as a potential design driver.

We did not take into account other load cases as our primary focus was on DLC 1.1. A designer still has to go through DLC 1.1 in order to discard it as a non-driving load case. Moreover, the results can also be applied to cases where loads are extrapolated from more realistic conditions with multiple variables (e.g., wind speed/wave height/atmospheric stability). In any case, the rather artificial, yet possibly design-driving DLC's in the IEC standards should not make the extrapolation problem less interesting.

5. Page 10, lines 16-20, the authors conclude: "In any case, we can conclude that the bare minimum of 300 minutes of time series, as prescribed in Appendix F of the standards (IEC, 2005), is not sufficient to produce any reasonable 50-year estimate. Based on this load set and this extrapolation produce, one should instead aim for sample sizes larger than $N = 10^5$." I do not agree with this conclusion. Due to the issues outlined in my comments above, some additional uncertainty is present in the extrapolation. If these uncertainties are eliminated, or if another extrapolation procedure (e.g. aggregation after fitting) is used, the accuracy of the extrapolation may improve and the required sample size will become smaller.

Apart from the approach used in the original manuscript, we have added some other approaches to strengthen the base for our conclusions (see point 1). We have also weakened our statement

on the 300 minutes a bit. Still, we can at least stick by our conclusion that 300 minutes is a very low minimum, especially considering modern-day computing power.

**Specific comments:**

6. Page 3, last paragraph: "the wind speeds belonging to the 10% highest loads points towards a region well above the rated wind speed (see Figure 3)". I don't think this statement agrees with Figure 3, where the mode of the distribution $f(\overline{U}|\hat{F} \geq 0.9)$ is actually at wind speeds just below rated.

The mode of the distribution $f(\overline{U}|\hat{F} \geq 0.9)$ is around 13.5 m/s, well above the rated wind speed of 11.4 m/s. In fact, if one were to take the top 1% or top 0.1% highest loads from Figure 1, they originate from increasingly higher wind speeds.

7. Page 4, line 5: Why is the GEV distribution a good candidate? In my experience, the GEV, or other 3- parameter representation as the 3-parameter Weibull, are good candidates for extrapolation in situations where only few data points are available and not all of them belong to the upper tail of the extremes distribution. However, if the amount of data and the threshold selection result in a data set which is predominantly from the upper tail which has a characteristic log-linear behaviour, I would think that a 2-parameter Gumbel distribution would provide more robust and accurate fit.

The GEV is a near perfect match to the tail for large sample sizes. Since characteristic log-linear behavior is not always found (for example the yawing moment in Barone, 2012), we wanted to avoid the issue of having to need prior knowledge about the tail. Nevertheless, the Gumbel distribution is usually more forgiving at small sample sizes. We have included it in the new version as an alternative method (see point 1).

8. Page 5, line 5: "any extrapolated 50-year loads that are more than 50% higher than the "real" value are discarded and resampled": - This data censoring approach is quite crude. It also relies on the known true value which will not be known in practice. Given the large number of samples, it is possible to establish a confidence interval for each load level and discard the outliers, see e.g. Naess and Gaidai, Structural Safety 31 (2009), pp. 325-334.

We have left the outliers in the new version to avoid having to know the true value.

9. Figure 9: It seems that the uncertainty for sample size N = 100 is smaller than the uncertainty for N = 1000. This is against the logic and indicates a possible sampling bias. Could the authors find an explanation for this?

This is because how the threshold is placed at very small sample sizes. For N < 100, the data above the threshold includes more of the knee in order to have enough points for a fit. The data points are then often lined up in a slightly downward curve, which favors a GEV with a negative ξ (thus

fewer outliers). For N > 1000, the threshold has moved further up the tail, which is close to being straight and favors both positive and negative $\xi$.

**Technical comments:**

10. Page 8, lines 5-10: It does not get entirely clear how the stress criteria for accepting or rejecting a given design relate to the extrapolated loads. Please improve the explanation.

The stress criteria was purely used as a simple example to demonstrate a decision making process. We have clarified this a little bit more in the new version.

---

## Author Comment (AC2) · 3 May 2017

Thank you for your comments and suggestions. They have certainly helped us to prepare a new version of the manuscript.

1. What is the procedure for obtaining the 50-year load from the data? There is some uncertainty associated with this value, and the authors should attempt to estimate it (for example, from a bootstrap procedure).

   The 50-year load is obtained from a GEV fit to the tail. We have clarified that in the new version. We have also estimated the 95% confidence level around the 50-year value by bootstrapping (13.1-17.2 MN m with a median at 115.0 MN m), and have added it in the new version.

2. It is not clear how the results might differ should one choose a different distribution other than GEV for the fit. While an exhaustive study of many possible distributions is not required, some discussion and exploration of this is in order

   In the new version, we have explored some different approaches to the extrapolation problem, including a Gumbel fit. This is more robust and forgiving at small sample sizes, but requires one to assume log-linear behavior in the tail.